# NaBH₄-Reduction Induced Evolution of Bi Nanoparticles from BiOCl Nanoplates and Construction of Promising Bi@BiOCl Hybrid Photocatalysts

**Yuxiang Yan [1], Hua Yang [1,\*] , Zao Yi [2] and Tao Xian [3]**

[1] State Key Laboratory of Advanced Processing and Recycling of Non-ferrous Metals, Lanzhou University of Technology, Lanzhou 730050, China; yanyx@lut.cn

[2] Joint Laboratory for Extreme Conditions Matter Properties, Southwest University of Science and Technology, Mianyang 621010, China; yizaomy@swust.edu.cn

[3] College of Physics and Electronic Information Engineering, Qinghai Normal University, Xining 810008, China; xiantao1985@126.com

[\*] Correspondence: hyang@lut.cn; Tel.: +86-931-297-3783

**Abstract:** In this work, we have synthesized BiOCl nanoplates (diameter 140–220 nm, thickness 60–70 nm) via a co-precipitation method, and then created Bi nanoparticles (diameter 35–50 nm) on the surface of BiOCl nanoplates via a NaBH₄ reduction method. By varying the NaBH₄ concentration and reaction time, the evolution of Bi nanoparticles was systematically investigated. It is demonstrated that with increasing the NaBH₄ concentration (at a fixing reaction time of 30 min), BiOCl crystals are gradually reduced into Bi nanoparticles, and pure Bi nanoparticles are formed at 120 mM NaBH₄ solution treatment. At low-concentration NaBH₄ solutions (e.g., 10 and 30 mM), with increasing the reaction time, BiOCl crystals are partially reduced into Bi nanoparticles, and then the Bi nanoparticles return to form BiOCl crystals. At high-concentration NaBH₄ solutions (e.g., 120 mM), BiOCl crystals are reduced to Bi nanoparticles completely with a short reaction time, and further prolong the treatment time leads to the transformation of the Bi nanoparticles into a two-phase mixture of BiOCl and Bi₂O₃ nanowires. The photodegradation performances of the samples were investigated by choosing rhodamine B (RhB) as the model pollutant and using simulated sunlight as the light source. It is demonstrated that an enhanced photodegradation performance can be achieved for the created Bi@BiOCl hybrid composites with appropriate NaBH₄ treatment. The underlying photocatalytic mechanism was systematically investigated and discussed.

**Keywords:** BiOCl nanoplates; Bi nanoparticles; NaBH₄ reduction; Bi@BiOCl hybrids; photodegradation performance

## 1. Introduction

The rapid development of chemical industry leads to the ever-increasing serious pollution of the human living environment. In particular, the industrial wastewater generated from textile/paper/paint/cosmetic manufacturers is causing severe pollution to the water resource. Organic dyes, being the dominant pollutants in the industrial wastewater, pose a great threat to mankind's health and aquatic life due to their harmful and carcinogenic properties [1]. Among various wastewater treatment technologies, semiconductor-based photocatalysis is particularly interesting because it has many outstanding advantages, such as utilization of sunlight as the power source, capability of decomposing organic dyes into harmless inorganic substances, environmental friendliness,

operation simplicity and low cost [2–6]. Under irradiation by the sun's power, semiconductor photocatalysts are excited to produce electrons (e⁻) in their conduction band (CB) and holes (h⁺) in their valence band (VB), and the photoexcited charge carriers are the original reactive species in the photodegradation process of dyes. To make more charge carriers available for the photocatalytic reactions and enhance the photocatalytic performance of semiconductors, the photoexcited electrons and holes must be efficiently separated [7–12].

In recent years, bismuth-based semiconductor photocatalysts have been extensively studied because of their excellent capabilities to photocatalytically degrade organic dyes [12–17]. Particularly, a great deal of recent interest has been paid to bismuth oxyhalides BiOX (X = I, Cl, Br), which are an important class of bismuth-based semiconductor photocatalysts. The BiOX compounds are constructed by the alternation between $[Bi_2O_2]^{2+}$ slabs and double layers of X atoms, thus resulting in the crystalline form of a layered PbFCl-type structure [18]. An interesting self-built internal static electric field (i.e., polarization electric field) is produced in the layer BiOX compounds [19]. The polarization electric field can promote the spatial separation of photogenerated electron/hole pairs, making BiOX exhibit a superior photocatalytic activity for the degradation of organic dyes. Moreover, much work has been devoted to the incorporation of BiOX with other semiconductors to form excellent heterojunction composite photocatalysts [20–24]. Due to the efficient separation of electron/hole pairs resulting from the charge transfer between the two semiconductors, those heterojunction composite photocatalysts generally exhibit photocatalytic performances superior to single semiconductor photocatalysts.

Noble metal nanoparticles and carbon nanomaterials (e.g., carbon quantum dots, carbon nanotubes and graphene) have been widely used to modify semiconductor photocatalysts with the aim of enhancing their photocatalytic performances [25–29]. Due to their intriguing physicochemical characteristics, the metal and carbon nanomaterials have potential technological applications in a wide range of fields such as electronic devices, biomedicine, sensors, and wave absorption [30–36]. In the aspect of photocatalytic applications, they can be used particularly as excellent electron captures to facilitate the separation of photoexcited electron/hole pairs. Furthermore, localized surface plasmon resonance (LSPR) could be induced in the metal nanoparticles by absorbing visible light [37–39]. The LSPR effect of metal nanoparticles can locally enhance the electromagnetic field, which facilitates the generation and separation of electron/hole pairs in semiconductor photocatalysts. These outstanding properties make metal nanoparticles particularly attractive as co-catalysts to improve the photocatalytic performances of semiconductor photocatalysts. Very recently, metal-modified BiOX composite photocatalysts have been extensively investigated, such as Au/BiOCl, Ag/BiOBr, Bi/BiOBr and Ag/Bi/BiOCl [40–46]. These composite photocatalysts were shown to exhibit superior performances for dye photodegradation when compared with bare BiOX. However, there is little work concerned with the evolution of Bi nanoparticles from BiOX crystals via a $NaBH_4$ reduction method and their effect on the optical and photocatalytic properties of resultant Bi@BiOX samples.

Herein, we presented a co-precipitation method for the synthesis of BiOCl nanoplates, and adopted a $NaBH_4$ reduction method to create Bi nanoparticles on the surface of BiOCl nanoplates. By varying the $NaBH_4$ solution concentration and treatment time, we systematically investigated the evolution process of Bi nanoparticles from BiOCl crystals. The photodegradation performances of the $NaBH_4$-treated BiOCl samples were assessed by degrading rhodamine B (RhB) from aqueous solution using simulated sunlight as the light source. RhB is one of the dominant organic dyes existing in the industrial wastewater. It must be artificially removed because it is highly water-soluble, chemically stable, carcinogenic, and non-biodegradable [47–49].

## 2. Results and Discussion

### 2.1. Effect of NaBH₄ Concentration

Figure 1 shows the XRD patterns of pristine and $NaBH_4$-treated BiOCl samples with different concentrations (reaction time 30 min). It is observed that the diffraction peaks of pristine BiOCl

sample matches perfectly with the standard diffraction pattern of PDF#85-0861, implying that BiOCl is crystallized into a pure P4/nmm tetragonal structure ($a$ = 0.389 nm, $b$ = 0.389 nm, $c$ = 0.737 nm). When the BiOCl sample is treated in $NaBH_4$ solution, additional diffraction peaks characterized as metallic Bi are observed on the XRD patterns. With increasing the concentration of $NaBH_4$ solution, the diffraction peak intensity of metallic Bi gradually increases, and a pure metallic phase Bi is formed when the $NaBH_4$ solution concentration is increased up to 120 mM. The Bi crystal has a R-3m rhombohedral structure (PDF#85-1329). The XRD result suggests that the $NaBH_4$ treatment leads to the gradual reduction of BiOCl into metallic Bi.

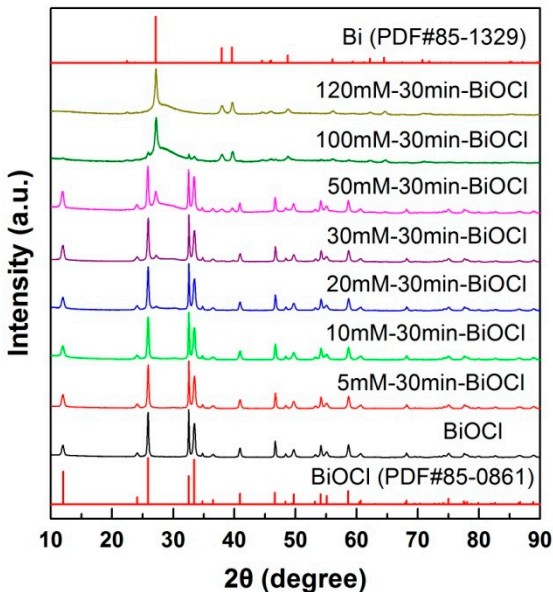

**Figure 1.** X-ray powder diffraction (XRD) patterns of pristine and $NaBH_4$-treated BiOCl samples with different concentrations (reaction time 30 min).

It is noted that the properties of nanomaterials are highly dependent on their optical absorption properties, which can be determined by UV-vis diffuse reflectance spectroscopy measurements [50–52]. Figure 2a demonstrates the apparent color change of the $NaBH_4$-treated BiOCl samples. Pristine BiOCl appears white in color, whereas the color of $NaBH_4$-treated BiOCl samples is gradually deepened to the black with increasing the $NaBH_4$ concentration. The color deepening can be explained due to the formation of Bi nanocrystals by the $NaBH_4$ reduction. It is well established that metal nanocrystals generally have a strong visible-light absorption due to the LSPR [53]. The enhanced visible-light absorption of the $NaBH_4$-treated BiOCl samples is further confirmed by the UV-vis diffuse reflectance spectra, as shown in Figure 2b. Owing to its large bandgap energy, pristine BiOCl manifests a poor optical absorption in the visible-light region. With increasing the NaBH4 concentration (particularly above 5 mM), the $NaBH_4$-treated BiOCl samples exhibit a monotonic increase in the visible-light absorption. Especially treated by 120 mM $NaBH_4$ solution, the obtained sample 120mM-30min-BiOCl absorbs fully visible light. To determine the bandgap energy ($E_g$) of the samples, the UV-vis diffuse reflectance spectra are differentiated, as shown in Figure 2c. According to the absorption edge (i.e., the peak wavelength [54]), the $E_g$ of the samples is obtained and presented in Figure 2c. The pristine BiOCl has an $E_g$ of 3.57 eV. A slightly decreased $E_g$ is observed for the $NaBH_4$-treated BiOCl samples, which could be ascribed to the lattice expansion of BiOCl crystals due to their partial reduction into Bi.

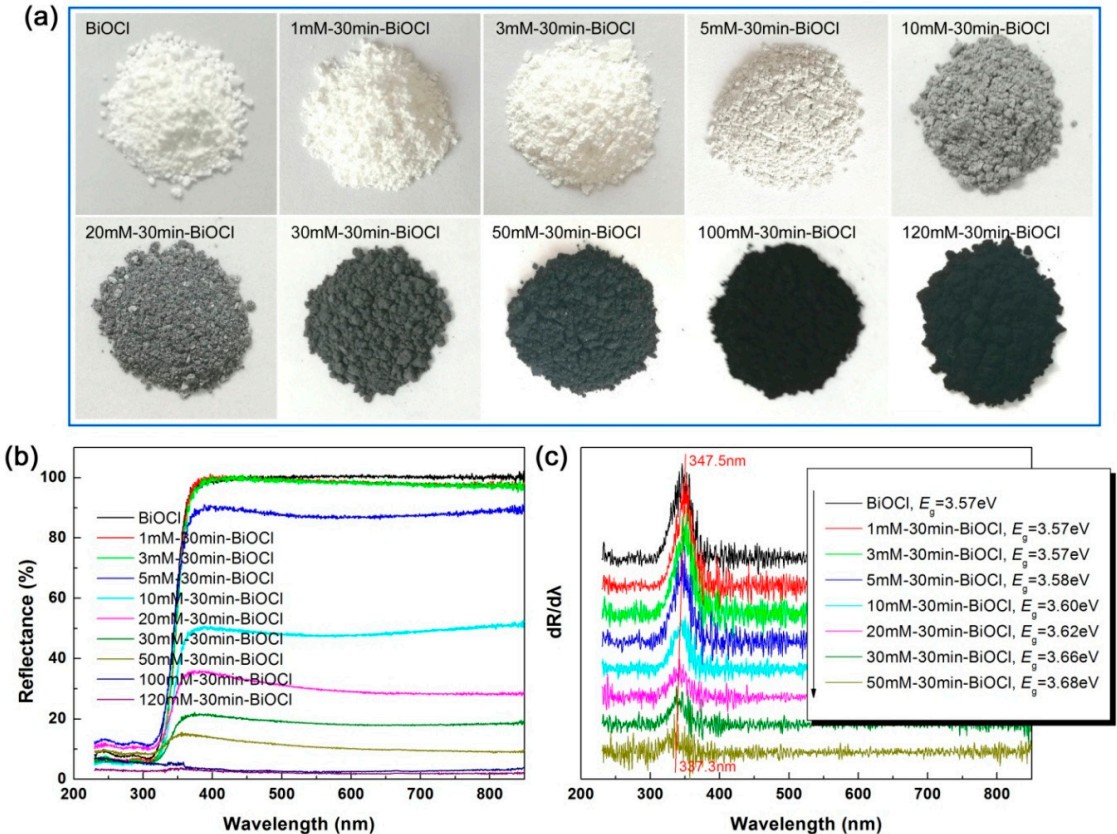

**Figure 2.** Apparent colors (**a**), UV-vis diffuse reflectance spectra (**b**), and first derivative curves of the diffuse reflectance spectra (**c**) of pristine and NaBH$_4$-treated BiOCl samples with different concentrations.

The morphologies of pristine and NaBH$_4$-treated BiOCl samples with different concentrations were elucidated by SEM observations. Figure 3a illustrates the SEM image of pristine BiOCl, revealing the synthesis of BiOCl nanoplates with size of 140–220 nm in diameter and 60–70 nm in thickness. Due to their large surface energy, the nanoplates are aggregated together to form rose-like microspheres. Figure 3b–f show the SEM images of NaBH$_4$-treated BiOCl samples with different concentrations. It is seen that with increasing the NaBH$_4$ solution concentration, BiOCl crystals are gradually reduced into Bi nanoparticles. Especially increasing the NaBH$_4$ concentration up to 120 mM, BiOCl crystals are completely decomposed and pure Bi nanoparticles are formed. The size of the Bi nanoparticles is about 35–50 nm.

TEM investigation was performed on pristine BiOCl, 30mM-30min-BiOCl and 120mM-30min-BiOCl to further unveil their microstructures. Figure 4a shows the TEM image of pristine BiOCl, revealing that it presents a morphology of nanoplates. The high-resolution TEM (HRTEM) image in Figure 4d shows that the BiOCl nanoplates are well crystallized and present clear lattice fringes with $d_{110}$ = 0.275 nm. The HRTEM image also shows the possible existence of subgrains in the BiOCl nanoplates. The selected area electron diffraction (SAED) pattern obtained from one of the BiOCl nanoplates is shown in Figure 4g, which clearly presents sharp diffraction spots. This is indicative of the good crystallization of the BiOCl nanoplates. However, the diffraction spots are arranged irregularly and nonperiodically, giving support to the possible existence of subgrains in the BiOCl nanoplates. For the 30mM-30min-BiOCl sample, its TEM image (Figure 4b) and HRTEM image (Figure 4e) demonstrate that partial BiOCl nanoplates are reduced by NaBH$_4$ to form Bi nanoparticles; while in the 120mM-30min-BiOCl sample, BiOCl nanoplates are completely transformed into Bi nanoparticles via the NaBH$_4$ reduction, as depicted from its TEM image (Figure 4c) and HRTEM image (Figure 4f). The Bi nanoparticles manifest obvious lattice fringes, which can be characterized as the rhombohedrally structured Bi. The HRTEM images also reveal the existence of minor amorphous Bi,

particularly at the surface of Bi crystals. This is confirmed by the SAED patterns of 30mM-30min-BiOCl (Figure 4h) and 120mM-30min-BiOCl (Figure 4i), both of which clearly show the diffuse diffraction rings that are characteristic of an amorphous material.

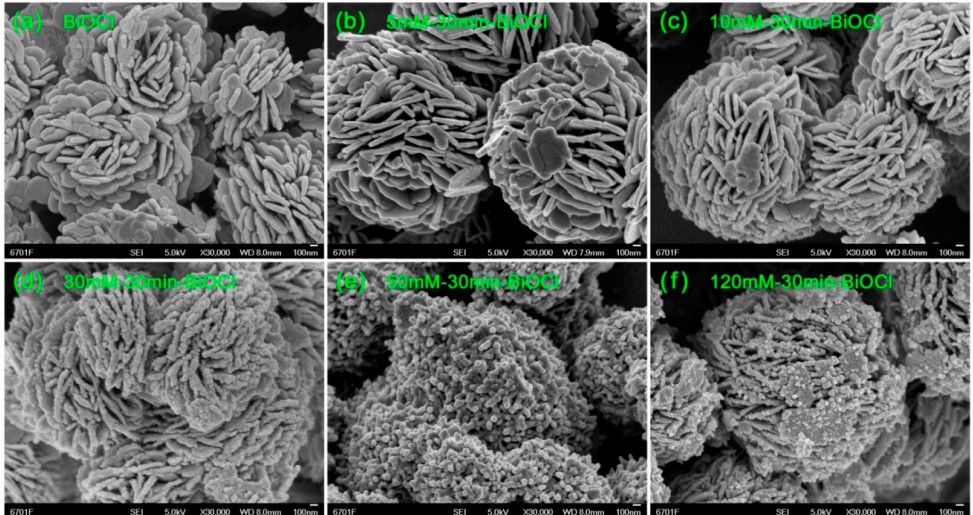

**Figure 3.** SEM images of pristine and NaBH$_4$-treated BiOCl samples with different concentrations.

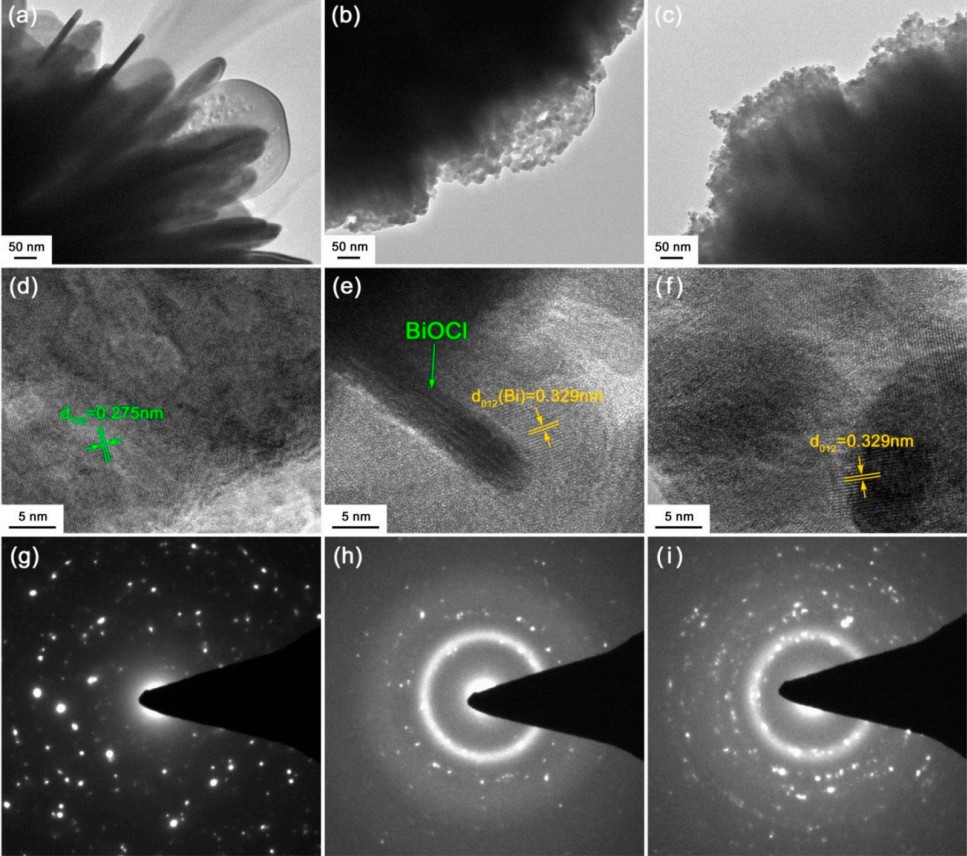

**Figure 4.** (**a**–**c**) TEM images of pristine BiOCl, 30mM-30min-BiOCl and 120mM-30min-BiOCl, respectively. (**d**–**f**) HRTEM images of pristine BiOCl, 30mM-30min-BiOCl and 120mM-30min-BiOCl, respectively. (**g**–**i**) SAED patterns of pristine BiOCl, 30mM-30min-BiOCl and 120mM-30min-BiOCl, respectively.

The element chemical states of 30mM-30min-BiOCl are analyzed by XPS spectra. Figure 5a shows the survey scan XPS spectrum of 30mM-30min-BiOCl, revealing that Bi, O and Cl elements are included in the sample. The C signal (C 1s → 284.8 eV) detected on the spectrum is used for the binding energy calibration [55]. Figure 5b–d illustrate the high-resolution XPS spectra of Bi 4f, O 1s and Cl 2p core levels, respectively. On the Bi 4f core-level spectrum, the two sharp peaks at 158.2 and 163.6 eV are attributed to Bi $4f_{7/2}$ and Bi $4f_{5/2}$ signals of $Bi^{3+}$ species, respectively [56]. The weak peaks at 157.0 and 162.3 eV are characterized as Bi $4f_{7/2}$ and Bi $4f_{5/2}$ signals of metallic $Bi^0$ species, implying the formation of Bi nanoparticles on the surface of BiOCl nanoplates [56]. The O 1s core-level spectrum presents two peaks at 529.0, and 530.5 eV, which are ascribed to the lattice oxygen of BiOCl crystals and chemisorbed oxygen species, respectively [56–58]. The existence of $Cl^-$ species is confirmed by the observation of two peaks at 197.1 (Cl $2p_{3/2}$) and 198.6 (Cl $2p_{1/2}$) on the Cl 2p core-level spectrum [56].

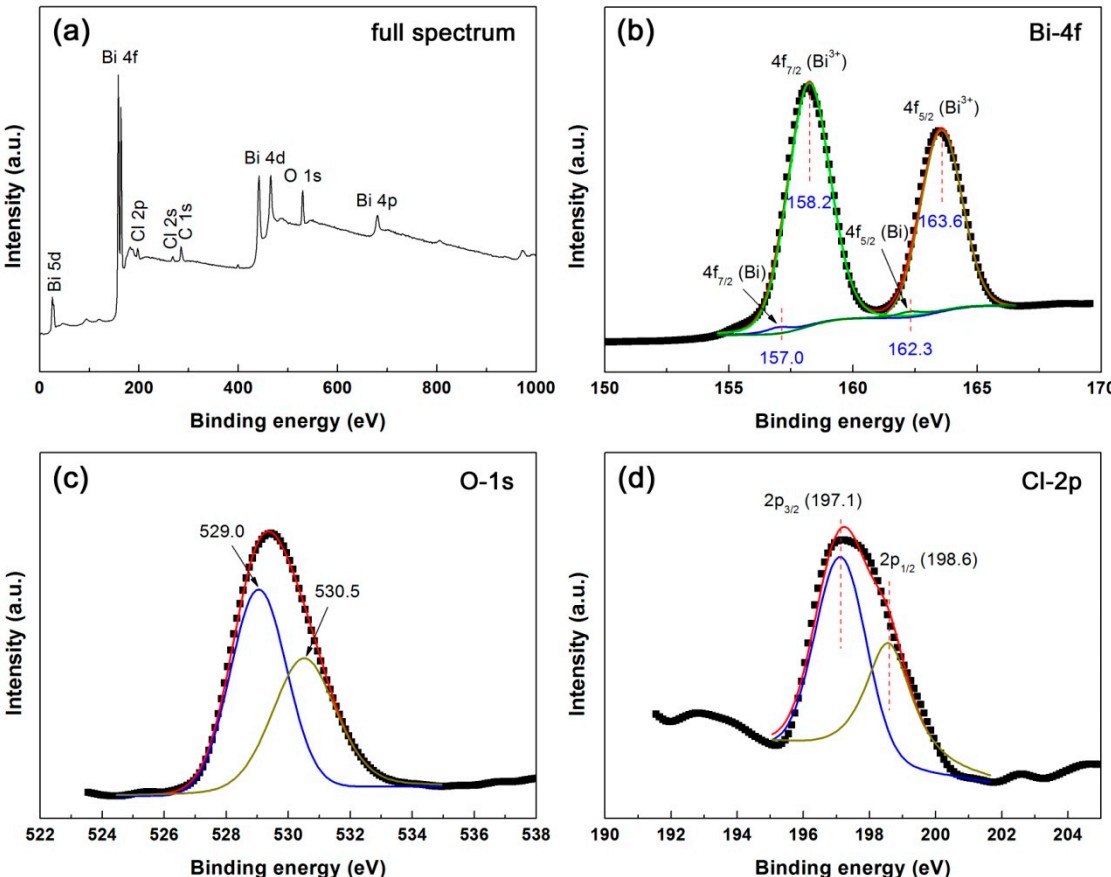

**Figure 5.** Survey scan XPS spectrum (**a**) and high-resolution XPS spectra of (**b**) Bi 4f, (**c**) O 1s and (**d**) Cl 2p of 30mM-30min-BiOCl.

Figure 6a shows the Raman spectra of BiOCl, 30mM-30min-BiOCl and 120mM-30min-BiOCl. The Raman spectrum of pristine BiOCl presents four peaks located at 56, 140, 196, and 394 $cm^{-1}$, which correspond to the $A_{1g}$ external Bi-Cl stretching mode, $A_{1g}$ internal Bi-Cl stretching mode, $E_g$ internal Bi-Cl stretching mode, and $E_g/B_{1g}$ mode produced by the motion of the oxygen atoms, respectively [59]. These characteristic peaks are indicative of the crystallization of BiOCl crystals. The characteristic Raman peaks of BiOCl crystals become sharply weakened for 30mM-30min-BiOCl, and completely disappear for 120mM-30min-BiOCl. Instead, three vibration bands at 86, 118 and 303 $cm^{-1}$ are detected on the Raman spectrum of 120mM-30min-BiOCl. These Raman bands are characterized as the vibration modes of metallic Bi [60], implying that treatment by 120 mM $NaBH_4$ solution results in the complete evolution of BiOCl crystals into metallic Bi nanoparticle.

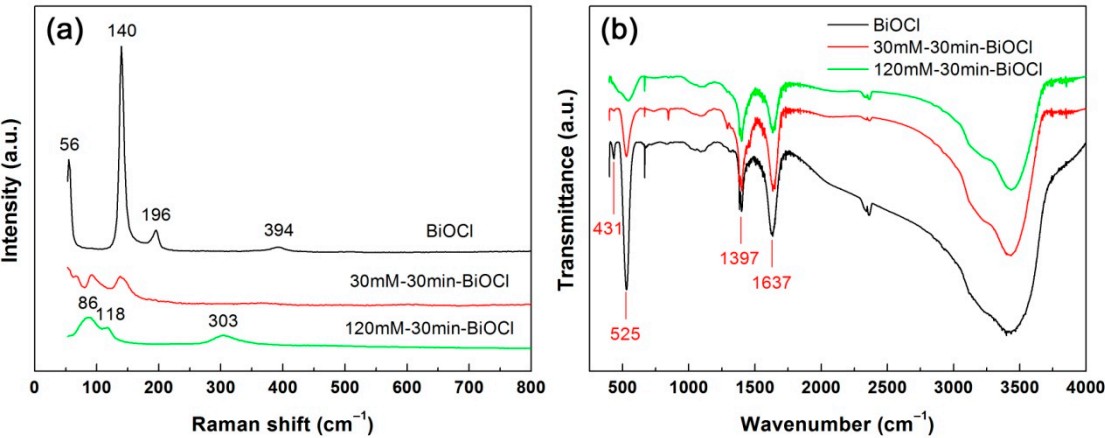

**Figure 6.** Raman spectra (**a**) and fourier transform infrared (FTIR) spectra (**b**) of pristine BiOCl, 30mM-30min-BiOCl and 120mM-30min-BiOCl.

FTIR is a useful technique for determining the functional groups. Figure 6b displays the FTIR spectra of BiOCl, 30mM-30min-BiOCl and 120mM-30min-BiOCl. A strong absorption peak at 525 cm$^{-1}$ and a weak absorption peak at 431 cm$^{-1}$ are observed for pristine BiOCl, which are characterized as the FTIR vibration peaks featured in BiOCl crystals [61]. The BiOCl characteristic peaks decrease in intensity for 30mM-30min-BiOCl, and disappear for 120mM-30min-BiOCl. This gives support to the fact that BiOCl crystals are completely reduced into Bi nanoparticles with 120 mM NaBH$_4$ solution treatment. No characteristic peaks from Bi nanoparticles are detected on the FTIR spectrum of 120mM-30min-BiOCl, possibly due to their infrared inactivity. The peak at 1397 cm$^{-1}$ could be ascribed to the in-plane deformation vibration of O–H groups [62], implying the possible existence of other organic impurities (e.g., alcohols) on the samples. The appearance of absorption peak at 1637 cm$^{-1}$ (H–O bending vibration) indicates that water molecules are adsorbed on the surface of the samples [63–65].

Photoelectrochemical measurements were carried out to elucidate the separation/transfer behavior of photoexcited carriers in the NaBH$_4$-treated BiOCl samples with different concentrations. Figure 7a shows the Nyquist plots of EIS data of the samples. It is seen that the Nyquist plots, except for that of 120mM-30min-BiOCl, display a typical semicircle in the high-frequency region and a straight line in the low-frequency region. With increasing the NaBH$_4$ concentration, the smallest semicircle diameter is observed for the 5mM-30min-BiOCl sample, implying that it has the smallest charge-transfer resistance [66]. The inset in Figure 7a illustrates the transient photocurrent-time curves, which are obtained by intermittently illuminating the samples with simulated sunlight. The samples manifest an obvious photocurrent response behavior with on-off cycles. On the irradiation, the obtained potocurrent density of the samples follows the order: 120mM-30min-BiOCl (0.05 μA cm$^{-2}$) < BiOCl (0.09 μA cm$^{-2}$) < 10mM-30min-BiOCl (0.12 μA cm$^{-2}$) < 3mM-30min-BiOCl (0.15 μA cm$^{-2}$) < 5mM-30min-BiOCl (0.16 μA cm$^{-2}$). Both the EIS and photocurrent response analyses demonstrate that NaBH$_4$-treated BiOCl samples with appropriate concentrations (particularly 5mM-30min-BiOCl) exhibit highly enhanced electron/hole separation and interface charge transfer.

PL spectroscopy is another important method that can be used to characterize the photoexcited carrier separation behavior. As seen in Figure 7b, the PL spectra of the NaBH$_4$-treated BiOCl samples show an obvious PL emission peak at 560 nm, which is induced by the recombination of photoexcited electron/hole pairs [67]. With increasing the NaBH$_4$ concentration, the 5mM-30min-BiOCl sample manifests the weakest PL emission peak, implying the lowest electron/hole recombination occurring in 5mM-30min-BiOCl. The PL spectroscopy analysis is consistent with the results obtained by the photocurrent response and EIS analyses.

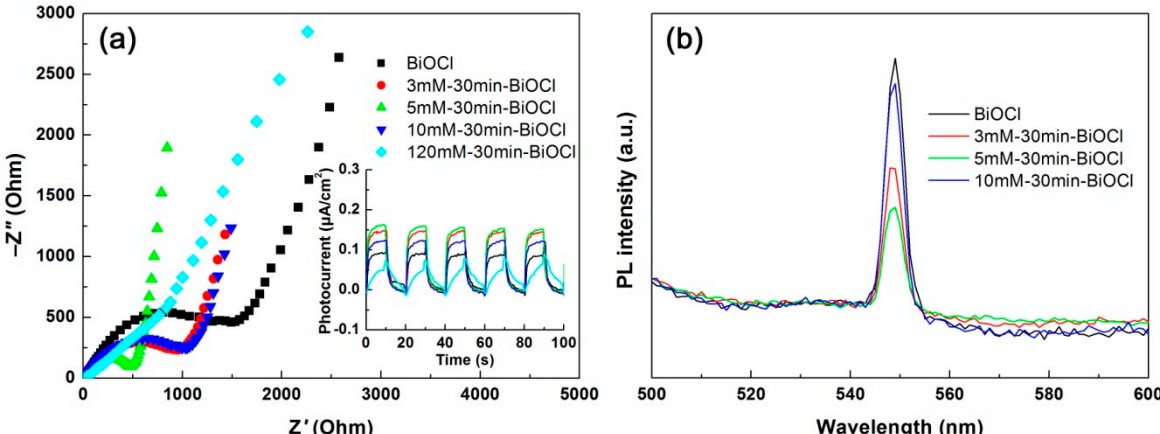

**Figure 7.** Nyquist plots of the electrochemical impedance spectroscopy (EIS) spectra (**a**), transient photocurrent response curves (inset in a), and PL spectra (**b**) of the NaBH$_4$-treated BiOCl samples with different concentrations.

The photodegradation performances of NaBH$_4$-treated BiOCl samples with different concentrations were assessed by eliminating RhB from aqueous solution under simulated sunlight irradiation. Figure 8a shows the adsorption and time-dependent photodegradation of RhB over the samples. The adsorption percentages (adsorption for 30 min in the dark) and degradation percentages of RhB (photoreaction for 30 min) are shown in Figure 8c. A large adsorption of RhB (33.9%) is observed on pristine BiOCl nanoplates, which could be induced by the polarization electric field existing in the BiOCl crystals [19]. The NaBH$_4$-treated BiOCl samples exhibit a decreased adsorption of RhB, which monotonically decreases with increasing the NaBH$_4$ concentration. This phenomenon can be explained by the gradual reduction of BiOCl crystals into Bi nanoparticles with increasing the NaBH$_4$ concentration, and Bi nanoparticles have a poor dye adsorption behavior. Pristine BiOCl nanoplates photocatalyze 81.3% removal of RhB after 30 min of irradiation. With increasing the NaBH$_4$ concentration, the photodegradation activity of the NaBH$_4$-treated BiOCl samples increases initially and then decreases. The highest degradation level of RhB reaches $\eta = 91.1\%$ for 5mM-30min-BiOCl. Only minor RhB is observed to be degraded for 120mM-30min-BiOCl (where BiOCl cystals are completely transformed to Bi nanoparticles), implying non-photoactivity of Bi nanoparticles. The degradation kinetic analysis is performed to further compare the photodegradation activity of the samples. As shown in Figure 8b, the plots of Ln($C_t/C_0$) *vs* $t$ for all the samples present a good linear behavior, and can be modeled using the pseudo-first-order kinetic equation Ln($C_t/C_0$) $= -k_{app}t$ [68]. $k_{app}$ is the apparent first-order reaction rate constant, which is obtained as given in Figure 8c. It is seen that the $k_{app}$ value increases from 0.05347 min$^{-1}$ for pristine BiOCl to 0.07945 min$^{-1}$ for the optimal NaBH$_4$-treated sample 5mM-30min-BiOCl, implying the latter has a photodegradation activity 1.5 times larger than that of the former.

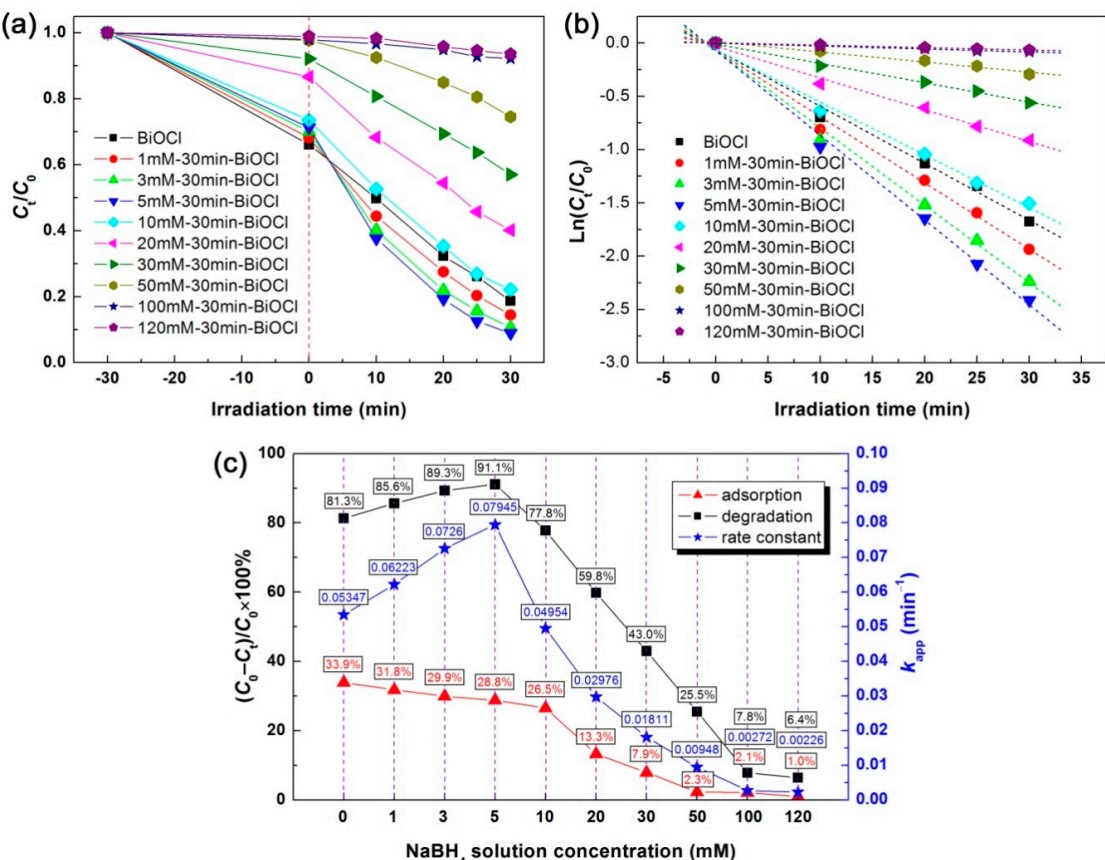

**Figure 8.** (**a**) Time-dependent photodegradation curves of RhB over the NaBH$_4$-treated BiOCl samples with different concentrations. (**b**) The corresponding kinetic plots of the dye degradation. (**c**) The adsorption/degradation percentages of RhB and the apparent first-order reaction rate constants of the dye degradation.

## 2.2. Effect of Treatment Time with NaBH$_4$

By fixing the NaBH$_4$ concentration separately at 10, 30 and 120 mM, we also investigated the effect of treatment time on the structure, optical property and photodegradation performance of NaBH$_4$-treated BiOCl samples. Figure 9a shows the apparent colors of the 10 mM NaBH$_4$-treated BiOCl samples with different reaction times. With increasing the reaction time, the color of the samples deepens from white (pristine BiOCl) to gray, and then returns to white (10mM-3h-BiOCl). The color change of the samples is confirmed by the UV-vis diffuse reflectance spectra, as shown in Figure 9b. The first derivative curves of the UV-vis diffuse reflectance spectra and the obtained $E_g$ values of the samples are shown in Figure 9c. The optical absorption properties suggest that, at the initial time, partial BiOCl crystals are transformed into Bi nanoparticles by NaBH$_4$ reduction. With further prolonging the treatment time, BiOCl crystals are recrystallized from the Bi nanoparticles. As only minor Bi nanoparticles are formed by 10 mM NaBH$_4$ treatment, the XRD patterns manifest no structural change, as shown in Figure 9d. The evolution between BiOCl crystals and Bi nanoparticles can be simply described by the following chemical reactions.

$$4BiOCl + 3NaBH_4 + 9H_2O \rightarrow 4Bi + 3B(OH)_3 + 3NaCl + HClO_4 + 10H_2 \tag{1}$$

$$4Bi + 3B(OH)_3 + 3NaCl + HClO_4 \rightarrow 4BiOCl + 3Na_3BO_3 + 5H_2 \tag{2}$$

The adsorption and photodegradation performances of the 10 mM NaBH$_4$-treated BiOCl samples with different reaction times are illustrated in Figure 9e, and the corresponding kinetic plots of the dye degradation are shown in Figure 9f. It is seen that the NaBH$_4$ treatment time has an obvious effect on the

adsorption and photodegradation performances of the samples. As discussed above, BiOCl nanoplates have a large adsorption and excellent photodegradation of RhB, whereas Bi nanoparticles have almost no adsorption and photodegradation performance. However, the incorporation of an appropriate number of Bi nanoparticles on the surface of BiOCl nanoplates could result in the enhanced photodegradation performance. On the other hand, when BiOCl nanoplates are treated with a high-concentration $NaBH_4$ solution and a long time, amorphous and defect structures will be produced in the samples, which is detrimental to the photodegradation performance. All these factors collectively lead to the effect of the treatment time on the adsorption and photodegradation performances of $NaBH_4$-treated BiOCl samples.

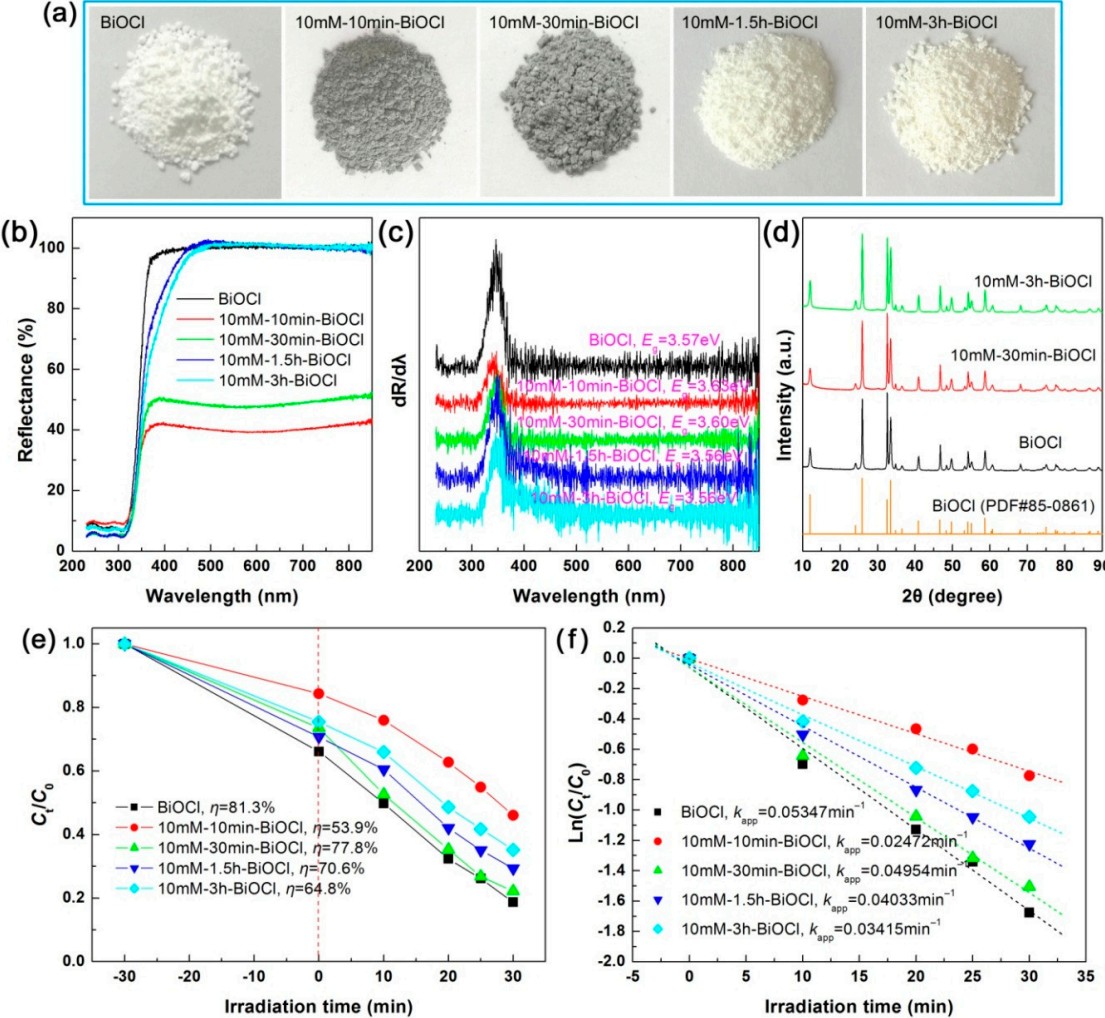

**Figure 9.** Apparent colors (**a**), UV-vis diffuse reflectance spectra (**b**), first derivative curves of the UV-vis diffuse reflectance spectra (**c**), XRD patterns (**d**), time-dependent photodegradation of RhB (**e**), and kinetic plots of the dye degradation (**f**) for the 10 mM NaBH4-treated BiOCl samples with different treatment times.

Figure 10a shows the apparent color change of the 30 mM $NaBH_4$-treated BiOCl samples with different treatment times. At a short-time treatment (10 min), the 30mM-10min-BiOCl sample becomes black in color. With increasing the treatment time, the color of the $NaBH_4$-treated BiOCl samples gradually becomes shallow, and finally reaches faint yellow for 30mM-6h-BiOCl. The UV-vis diffuse reflectance spectra illustrated in Figure 10b confirm the variation of the visible-light absorption properties of the samples. The $E_g$ values of the samples, derived from the first derivative curves of the UV-vis diffuse reflectance spectra (Figure 9c), manifest a slight difference between the samples,

which could be ascribed to the interaction between Bi nanoparticles and BiOCl crystals. With increasing the treatment time, the evolution process from BiOCl crystals partially to Bi nanoparticles, and then from Bi nanoparticles to BiOCl crystals can be verified by the XRD patterns, as shown in Figure 9d. In comparison with pristine BiOCl, weak diffraction peaks of metallic Bi are additionally detected for 30mM-30min-BiOCl; whereas the Bi diffraction peaks disappear for 30mM-6h-BiOCl. In addition to a weak amorphous peak appearing at around 2θ = 30°, all the diffraction peaks of 30mM-6h-BiOCl can be perfectly indexed to the BiOCl phase. As seen from the SEM image in Figure 10e, the 30mM-6h-BiOCl sample presents a morphology of nanowires/nanoparticles, which is different from that of pristine BiOCl (Figure 3a).

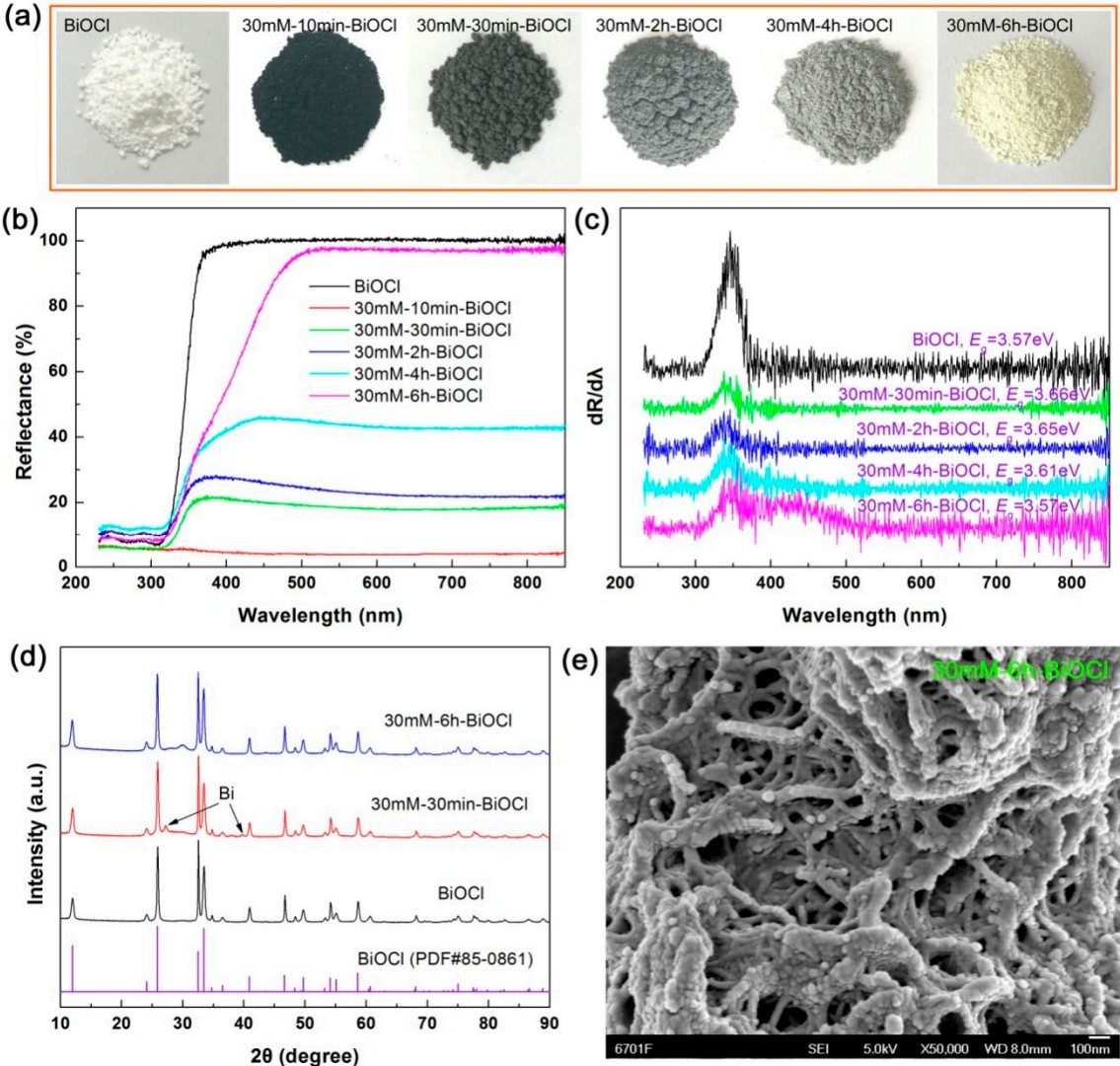

**Figure 10.** (**a**–**d**) Apparent colors, UV-vis diffuse reflectance spectra, first derivative curves of the UV-vis diffuse reflectance spectra, and XRD patterns of the 30 mM NaBH$_4$-treated BiOCl samples with different treatment times, respectively. (**e**) SEM image of 30mM-6h-BiOCl.

Figure 11a shows the photodegradation performances of the 30 mM NaBH$_4$-treated BiOCl samples with different treatment times, and Figure 11b displays the corresponding kinetic plots of the dye degradation. It is seen that the 30mM-6h-BiOCl sample manifests the smallest adsorption and photodegradation toward RhB, which is ascribed to the fact that excess Bi nanoparticles are evolved from BiOCl crystals by the NaBH$_4$ reduction. With increasing the treatment time, the Bi nanoparticles gradually return into BiOCl crystals, and the optimal Bi@BiOCl hybrid composite with an

appropriate ratio between Bi and BiOCl could be formed. However, amorphous and defect structures are simultaneously produced. As a result, the optimal NaBH₄-treated sample—30mM-4h-BiOCl still exhibits a photodegradation activity smaller than that of pristine BiOCl nanoplates.

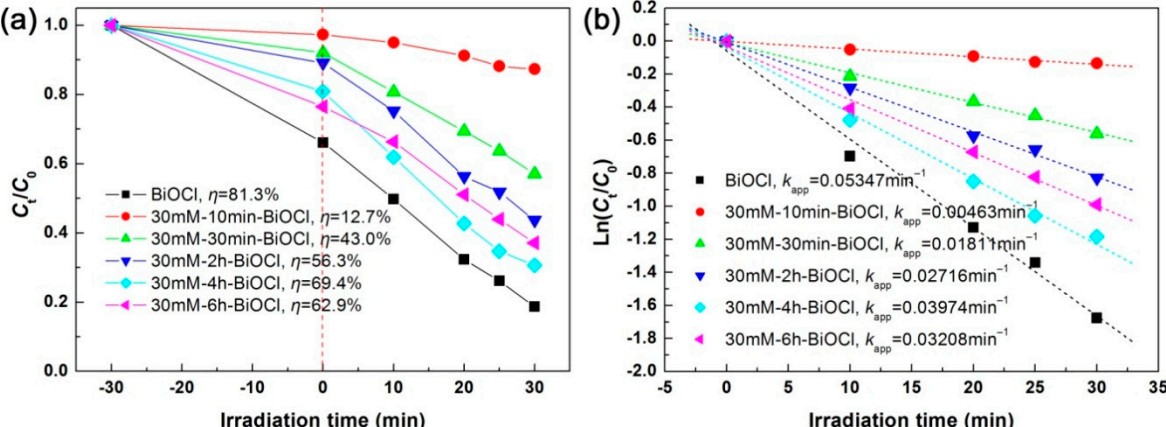

**Figure 11.** Time-dependent photodegradation of RhB (**a**) and kinetic plots of the dye degradation (**b**) over the 30 mM NaBH₄-treated BiOCl samples with different treatment times.

Figure 12a–c show the apparent colors, UV-vis diffuse reflectance spectra, and first derivative curves of the UV-vis diffuse reflectance spectra of the 120 mM NaBH₄-treated BiOCl samples with different treatment times, respectively. It is seen that the samples treated at short times have extremely strong visible-light absorption. When the treatment time exceeds 5 h, the samples exhibit a gradual decrease in the visible-light absorption. The optical absorption change of the BiOCl samples with different NaBH₄ treatment times is highly dependent on their structural evolution, as elucidated by XRD investigation (Figure 12d). Under high-concentration NaBH₄ solution treatment (120 mM), BiOCl crystals are reduced to Bi nanoparticles completely with a short time. With prolonging the treatment time, the Bi crystals are dissolved and recrystallized into a two-phase mixture of BiOCl and $Bi_2O_3$. On the first derivative UV-vis diffuse reflectance spectra of 120mM-5h-BiOCl and 120mM-9h-BiOCl (Figure 12c), the detected absorption edge at 421.2 nm can be attributed to $Bi_2O_3$. The SEM image of 120mM-9h-BiOCl (Figure 12e) demonstrates that BiOCl and $Bi_2O_3$ have a morphology of nanowires.

Figure 13a illustrates the photodegradation and adsorption performances of the 120 mM NaBH₄-treated BiOCl samples with different treatment times toward RhB, and Figure 13b gives the corresponding kinetic plots of the dye degradation. It is seen that the samples treated at short times (below 2 h) have extremely small adsorption and photodegradation toward RhB, which can be explained due to the full evolution of BiOCl nanoplates into Bi nanoparticles. Further prolonging the treatment time leads to the transformation of Bi nanoparticles into a two-phase mixture of BiOCl and $Bi_2O_3$, and furthermore $BiOCl/Bi_2O_3$ heterostructures could be formed, which are beneficial to the dye degradation. On the other hand, the formed amorphous and defect structures are detrimental to the photodegradation performance. Among the 120 mM NaBH₄-treated BiOCl samples, the highest photodegradation activity is observed for 120mM-9h-BiOCl, but it is smaller than that of pristine BiOCl nanoplates.

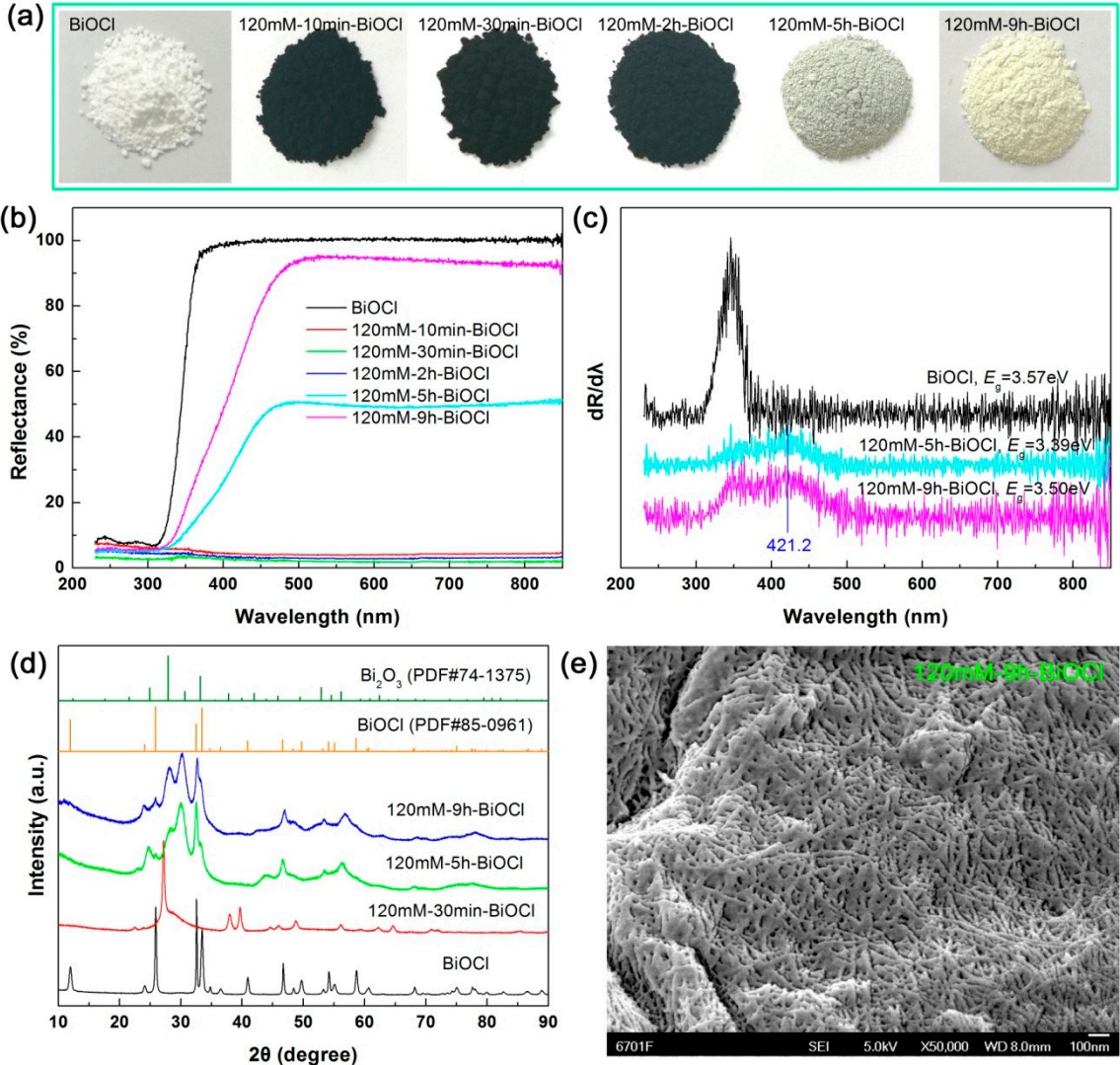

**Figure 12.** (**a–d**) apparent colors, UV-vis diffuse reflectance spectra, first derivative curves of the UV-vis diffuse reflectance spectra, and XRD patterns of the 120 mM NaBH$_4$-treated BiOCl samples with different treatment times, respectively. (**e**) SEM image of 120mM-9h-BiOCl.

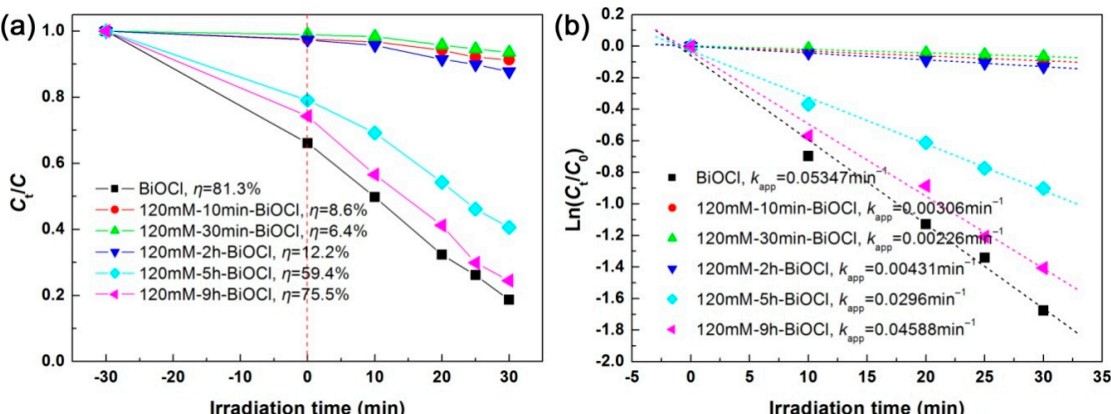

**Figure 13.** Time-dependent photodegradation of RhB (**a**) and kinetic plots of the dye degradation (**b**) over the 120 mM NaBH$_4$-treated BiOCl samples with different treatment times.

### 2.3. Photodegradation Mechanism of Bi@BiOCl Hybrid Photocatalysts

The above experimental results and analyses suggest that the photodegradation performance of BiOCl nanoplates can be enhanced by the NaBH$_4$ solution treatment (with an appropriate NaBH$_4$ concentration and at a proper treatment time). This phenomenon can be explained due to the creation of Bi nanoparticles on the surface of BiOCl nanoplates, as schematically depicted in Figure 14a. The enhanced photodegradation mechanism of the Bi@BiOCl hybrid composites is schematically illustrated in Figure 14b. When the Bi@BiOCl hybrid composites, both BiOCl nanoplates and Bi nanoparticles are excited. The photoexcitation of BiOCl nanoplates leads to the generation of electrons in their CB and holes in their VB. The photoexcitation of Bi nanoparticles is ascribed to the localized surface plasmon resonance. It is well established the excited metal nanoparticles can act as excellent electron sinks [69], and as a result, the photoexcited electrons in the CB of BiOCl will be transferred to Bi nanoparticles. Simultaneously, the LSPR-induced electrons in Bi nanoparticles could be also transferred to the CB of BiOCl, as depicted in Figure 14b. This electron transfer process can facilitate the separation of photoexcited e$^-$/h$^+$ pairs in BiOCl nanoplates, which is confirmed by the EIS, photocurrent response and PL spectroscopy analyses (Figure 7). More holes in the VB of BiOCl are therefore able to participate in the photocatalytic reactions. The LSPR-induced electrons in Bi nanoparticles could also play a role in the dye degradation. Furthermore, the LSPR of Bi nanoparticles induce locally enhanced electromagnetic field, which will stimulate the generation and separation of e$^-$/h$^+$ pairs in BiOCl. Due to these factors, the formed Bi@BiOCl hybrid composites with appropriate NaBH$_4$ treatment manifest enhanced photodegradation of RhB under simulated sunlight irradiation.

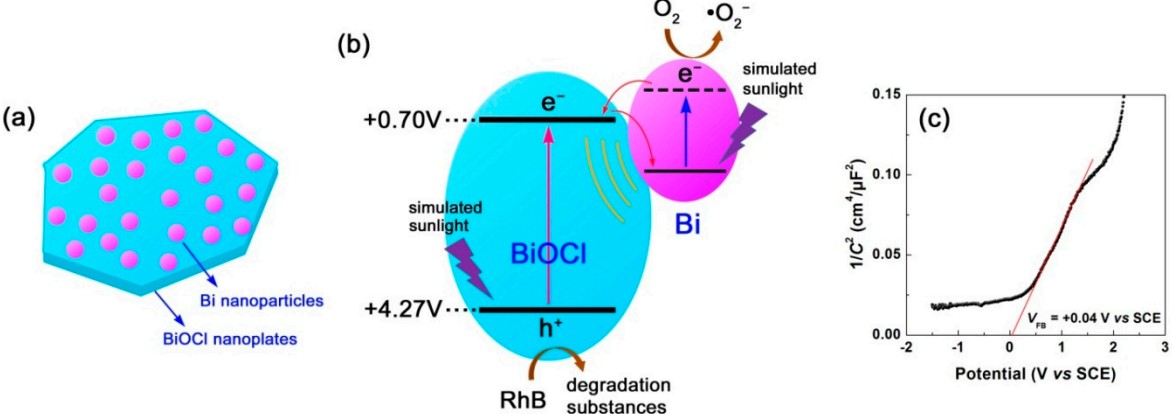

**Figure 14.** (**a**) Schematic illustration of the Bi@BiOCl hybrid composites with Bi nanoparticles assembled on the surface of BiOCl nanoplates. (**b**) Schematic illustration of the photodegradation mechanism of the Bi@BiOCl hybrid composites. (**c**) Mott–Schottky plot of BiOCl nanoplates.

To further understand the photodegradation mechanism of the Bi@BiOCl hybrid composites, Mott–Schottky method [70] was used to determine the CB and VB potentials of BiOCl nanoplates. Figure 14c shows the Mott–Schottky plot of BiOCl derived from the electrochemical measurement at 3000 Hz. By extrapolating the linear portion of the plot to the x-axis, the flat band potential ($V_{FB}$) of BiOCl is obtained as $V_{FB}$(SCE) = +0.04 V vs. SCE, correspondingly $V_{FB}$(NHE) = +0.70 V vs. normal hydrogen electrode (NHE) according to $V$(NHE) = $V$(SCE) + 0.059 × pH(=7) + 0.242 [17]. BiOCl behaves as a n-type semiconductivity due to the positive slope of the Mott–Schottky plot, and its CB edge potential can be approximately equal to the value of its flat band potential. Thus, the CB and VB potentials of BiOCl nanoplates ($E_g$ = 3.57 eV) are obtained as +0.70 and +4.27 V vs. NHE, respectively.

Generally, in a photocatalytic system, the dominant reactive species causing the dye degradation include photoexcited holes, hydroxyl (●OH) radicals and superoxide (●O$_2^-$) radicals. In the present Bi@BiOCl photocatalytic system, the role of ●OH radicals can be negligible according to the reactive species trapping experiments performed using the method described in our previous work [14].

Whereas photoexcited holes and $\bullet O_2{}^-$ radicals are confirmed to be the main reactive species causing the dye degradation. Although the VB potential of BiOCl (+4.27 V vs. NHE) is sufficiently positive compared to $E^0(H_2O/\bullet OH)$ = +2.38 vs. NHE and $E^0(OH^-/\bullet OH)$ = +1.99 vs. NHE [14], the photoexcited holes do not react with $OH^-$ or $H_2O$ to produce $\bullet OH$, instead directly oxidize the dye. One of the feasible ways to generate $\bullet O_2{}^-$ radicals is through the reaction of adsorbed $O_2$ molecules with the LSPR-induced electrons in Bi nanoparticles. Moreover, we cannot exclude the possibility that the photoexcited electrons at higher excited states of BiOCl could also react with $O_2$ to produce $\bullet O_2{}^-$, though the CB potential of BiOCl (+0.70 V vs. NHE) is not negative to $E^0(O_2/\bullet O_2{}^-)$ = −0.13 V vs. NHE [14].

## 3. Materials and Methods

### 3.1. Materials

The raw materials and chemical reagents (analytical grade) purchased from manufacturers were directly used in the experiments without further purification. Bismuth nitrate pentahydrate $(Bi(NO_3)_3 \cdot 5H_2O, \geq 99\%)$ and sodium borohydride $(NaBH_4, \geq 99\%)$ were obtained from Shandong Xiya Chemical Industry Co., Ltd. Glacial acetic acid $(CH_3COOH, \geq 99.5\%)$ and sodium chloride $(NaCl, \geq 99.5\%)$ were purchased from Guangdong Chemical Reagent Engineering-Technological Research and Development Center. Polyvinyl pyrrolidone (PVP) was purchased from Shanghai Aladdin Reagent Co., Ltd. Methanol $(CH_3OH, \geq 99.5\%)$ was derived from Tianjin Baishi Chemical Industry Co. Ltd.

### 3.2. Synthesis of BiOCl Nanoplates

In a typical synthesis of BiOCl nanoplates based on a co-precipitation route, 1.4553 g (3 mMol) of $Bi(NO_3)_3 \cdot 5H_2O$ and 6 mL of glacial acetic acid were dissolved in 30 mL deionized water with magnetic stirring for 1 h (designated as solution A). 0.1753 g (3 mMol) of NaCl was added in 30 mL methanol, followed by 1 h magnetic stirring (designated as solution B). The solution B was slowly added in the solution A drop by drop, followed by magnetic stirring for 12 h. The precipitate was washed with deionized water (3 times) and ethanol (3 times) to remove impurity ions. After dried at 60 °C for 12 h, the product was obtained as BiOCl nanoplates.

### 3.3. Treatment of BiOCl Nanoplates with $NaBH_4$

The as-synthesized BiOCl nanoplates were treated with $NaBH_4$ solution. The BiOCl sample treated with $x$ mMol $L^{-1}$ (mM) $NaBH_4$ solution for $t$ min was termed as $x$mM-$t$min-BiOCl. For example, to obtain 10mM-30min-BiOCl with $NaBH_4$ solution treatment, 1 g of PVP was added in 100 mL deionized water with 1 h magnetic stirring, and then 0.5029 g (2 mMol) of BiOCl nanoplates was added in the PVP solution with another 1 h magnetic stirring. There was 30 mL of 10 mM $NaBH_4$ solution slowly added in the above suspension drop by drop, followed by 30 min of magnetic stirring (i.e., reaction for 30 min). After washed with deionized water and ethanol, and dried at 60 °C for 12 h, the final treated sample was obtained as 10mM-30min-BiOCl.

### 3.4. Characterization Methods

A D8 Advance X-ray diffractometer (Bruker AXS, Karlsruhe, Germany) with $\lambda_{Cu-K\alpha}$ = 0.15406 nm was used for the X-ray powder diffraction (XRD) characterization of the samples. Ultraviolet-visible (UV-vis) diffuse reflectance spectroscopy measurements were carried out on a TU-1901 double beam UV-vis spectrophotometer (Beijing Purkinje General Instrument Co. Ltd., Beijing, China). The scanning/transmission electron microscopy (SEM/TEM) observations were performed on a JSM-6701F field-emission scanning electron microscope (JEOL Ltd., Tokyo, Japan) and a JEM-1200EX field-emission transmission electron microscope (JEOL Ltd., Tokyo, Japan). X-ray photoelectron spectroscopy (XPS) measurement was carried out on a PHI-5702 multi-functional X-ray photoelectron

spectrometer (Physical Electronics, hanhassen, MN, USA). Fourier transform infrared (FTIR) spectra were obtained on a Spectrum Two FTIR spectrophotometer (PerkinElmer, Waltham, MA, USA). A LabRAM HR Evolution Raman spectrometer (Horiba Jobin Yvon, France) was used for the Raman microscopy measurements. A RF-6000 fluorescence spectrophotometer (Shimadzu, Kyoto, Japan) was employed to measure the photoluminescence (PL) spectra at an excitation wavelength of 325 nm. All characterizations were performed on the powder samples as-prepared.

Photocurrent response and electrochemical impedance spectroscopy (EIS) were measured on a CST 350 electrochemical workstation (Wuhan Corrtest Instruments Co. Ltd.,Wuhan, China). A three-electrode cell configuration, where a standard calomel electrode (SCE) acted as the reference electrode and a platinum foil acted as the counter-electrode, was used during the photoelectrochemical measurements. The working electrode was prepared according the procedure as described in the literature [71]. The used electrolyte was 0.1 M $Na_2SO_4$ aqueous solution, and the light source was a 200-W xenon lamp that emits simulated sunlight (wavelength region: 300–2500 nm).

*3.5. Photocatalytic Testing*

The photodegradation performances of $NaBH_4$-treated BiOCl samples were investigated by removing RhB from aqueous solution under illumination by a 200-W xenon lamp (a good sunlight simulator [72]). The sunlight simulator has a wavelength region of 300–2500 nm and total irradiance of $0.58\,W\,cm^{-1}$ at a distance of 1 cm. 0.03 g of photocatalyst and 100 mL of RhB solution ($C_{photocatalyst} = 0.3\,g\,L^{-1}$, $C_{RhB} = 5\,mg\,L^{-1}$) were loaded in the photoreactor. Before the photodegradation experiment, the mixture was magnetically stirred in the dark for 30 min with the aim of determining the adsorption of RhB onto the photocatalyst. The residual RhB concentration of the reaction solution during the photodegradation process was monitored by measuring its absorbance. To achieve this aim, 2.5 mL of the reaction solution was sampled from the photoreactor and then centrifuged (4000 rpm, 10 min) to separate the photocatalyst. A UV-vis spectrophotometer was used to measure the absorbance of the reaction solution at a given wavelength $\lambda$ = 554 nm. Based on the initial concentration ($C_0$) and residual concentration ($C_t$) of RhB, the degradation percentage ($\eta$) of RhB was given as: $\eta = (C_0 - C_t)/C_0 \times 100\%$.

## 4. Conclusions

Herein, the $NaBH_4$-reduction induced evolution of Bi nanoparticles from BiOCl nanoplates was systematically investigated by varying the $NaBH_4$ concentration and reaction time. With increasing the $NaBH_4$ concentration from 1 to 120 mM (reaction time 30 min), BiOCl crystals are gradually transformed into Bi nanoparticle by the $NaBH_4$ reduction. At low-concentration $NaBH_4$ solution (e.g., 10 and 30 mM), the increase in the reaction time leads to the transformation firstly from partial BiOCl crystals into Bi nanoparticles, and then from the Bi nanoparticles into BiOCl crystals. At high-concentration $NaBH_4$ solutions (e.g., 120 mM) and with increasing the reaction time, BiOCl crystals are completely reduced to Bi nanoparticles at a short reaction time, and then the Bi nanoparticles are gradually transformed into a two-phase mixture of BiOCl and $Bi_2O_3$ nanowires. Photodegradation experiments under simulated sunlight illumination demonstrate that the Bi@BiOCl hybrid composites created by appropriate $NaBH_4$ treatment can achieve an excellent photodegradation performance, which is higher than that of pristine BiOCl nanoplates. The enhanced photocatalytic mechanism can be explained due to the three aspects: (1) The electron transfer process from the CB of BiOCl to the LSPR-excited Bi nanoparticles promotes the separation of photoexcited $e^-/h^+$ pairs in BiOCl nanoplates; (2) The LSPR-induced electrons in Bi nanoparticles could also participate in the photodegradation reactions; (3) The LSPR-induced electromagnetic field from Bi nanoparticles could facilitate the generation and separation of $e^-/h^+$ pairs in BiOCl nanoplates.

**Author Contributions:** H.Y. conceived the idea of experiment; Y.Y. performed the experiments; H.Y., Y.Y., Z.Y. and T.X. discussed the results; H.Y. wrote the manuscript; All authors read and approved the final manuscript.

**Funding:** This research was funded by the National Natural Science Foundation of China (Grant No. 51662027) and the HongLiu First-Class Disciplines Development Program of Lanzhou University of Technology.

**Conflicts of Interest:** The authors declare that they have no competing interest.

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
