# Peer review of "NaBH4-Reduction Induced Evolution of Bi Nanoparticles from BiOCl Nanoplates and Construction of Promising Bi@BiOCl Hybrid Photocatalysts"

_catalysts, doi:10.3390/catal9100795_

Round 1

Reviewer 1 Report

Manuscript ID: catalysts-603514

Title: NaBH4-reduction induced evolution of Bi nanoparticles from BiOCl nanoplates and construction of promising Bi@BiOCl hybrid photocatalysts.

Reviewer comments:

The authors have produced an interesting and complete study concerning the preparation of Bi nanoparticles from BiOCl nanoplates, varying NaBH4 concentration and reaction time, for photocatalytic applications. The work has been undertaken diligently, well-written and the interpretation of the data supports the conclusions. However, some points should be modified. I would suggest publishing it on Catalysts after minor revisions.

(1) Introduction: The choice of rhodamine B (RhB) as model pollutant should be briefly explained, adding relevant references (for example https://doi.org/10.1021/la904023j; https://doi.org/10.1021/j100534a012; https://doi.org/10.1016/j.jenvman.2006.11.002)

(2) Materials&Methods: The authors should explain the morphology of the samples used for SEM/TEM/FTIR analyses (powders? Foils?) because the performance of such kind of techniques is straightly dependent from samples morphology.

(3) Photocatalytic testing: the authors should add relevant data about photocatalytic test (effective power density of the lamp, distance between sample and lamp, chamber T) and explain the choice of such kind of lamp. For example, looking in Catalysts this recent paper https://doi.org/10.3390/catal8110568, photocatalytic tests are conducted comparing data obtained with UV and solar simulator lamps.

(4) Effect of treatment time with NaBH4: The authors should explain if the modulation of device treatment time with NaBH4 could allow obtaining a recyclable device.

Author Response

Response to Reviewer#1

Response: We thank very much the reviewer for carefully reading out manuscript and giving us valuable comments and suggestions. According to the reviewer’s comments and suggestions, we have revised our manuscript and the revised parts are highlighted by red color.

The authors have produced an interesting and complete study concerning the preparation of Bi nanoparticles from BiOCl nanoplates, varying NaBH4 concentration and reaction time, for photocatalytic applications. The work has been undertaken diligently, well-written and the interpretation of the data supports the conclusions. However, some points should be modified. I would suggest publishing it on Catalysts after minor revisions.

(1) Introduction: The choice of rhodamine B (RhB) as model pollutant should be briefly explained, adding relevant references (for example https://doi.org/10.1021/la904023j; https://doi.org/10.1021/j100534a012; https://doi.org/10.1016/j.jenvman.2006.11.002)

Response: Thank the reviewer for giving this good comment. According to the reviewer’s suggestion, we have briefly explained the reason why RhB was chosen as model pollutant to assess the photodegradation activity of the samples. The papers mentioned by the reviewer have been cited.

(2) Materials&Methods: The authors should explain the morphology of the samples used for SEM/TEM/FTIR analyses (powders? Foils?) because the performance of such kind of techniques is straightly dependent from samples morphology.

Response: Thank the reviewer for giving this good comment. All characterizations were performed on the powder samples as-prepared.

(3) Photocatalytic testing: the authors should add relevant data about photocatalytic test (effective power density of the lamp, distance between sample and lamp, chamber T) and explain the choice of such kind of lamp. For example, looking in Catalysts this recent paper https://doi.org/10.3390/catal8110568, photocatalytic tests are conducted comparing data obtained with UV and solar simulator lamps.

Response: This suggestion is very good. According to the reviewer’s suggestion, we have added the relevant data about photocatalytic test. A 200-W xenon lamp was used as the light source because it is a good sunlight simulator. The paper mentioned by the reviewer has been cited.

(4) Effect of treatment time with NaBH4: The authors should explain if the modulation of device treatment time with NaBH4 could allow obtaining a recyclable device.

Response: Thank the reviewer for giving this good comment. In the following research, we will consider the design a recyclable device.

Reviewer 2 Report

This work describes the use of a hybrid photocatalyst based on Bi nanoparticles obtained by the reduction of NaBH4 from BiOCl nanoplates. The organic dye Rhodamine B was used as a reference compound for photodegradation experiments. The degradation kinetics of the catalyst were evaluated by varying the concentration of NaBH4 solution and the reaction time, and the Bi@BiOCl hybrids exhibit an enhanced photodegradation performance, compared to BiOCl nanoplates. The experimental part is well described, and several characterization methods were discussed in detail. The manuscript is suitable for publication as itself. My only suggestion is to specify the acronyms.

Author Response

Response to Reviewer#2

This work describes the use of a hybrid photocatalyst based on Bi nanoparticles obtained by the reduction of NaBH4 from BiOCl nanoplates. The organic dye Rhodamine B was used as a reference compound for photodegradation experiments. The degradation kinetics of the catalyst were evaluated by varying the concentration of NaBH4 solution and the reaction time, and the Bi@BiOCl hybrids exhibit an enhanced photodegradation performance, compared to BiOCl nanoplates. The experimental part is well described, and several characterization methods were discussed in detail. The manuscript is suitable for publication as itself. My only suggestion is to specify the acronyms.

Response: We thank very much the reviewer for carefully reading out manuscript and giving a positive comment. We have carefully checked and specified the acronyms according to the reviewer’s suggestion.
